# Patient Dietary Supplements Use: Do Results from Natural Language Processing of Clinical Notes Agree with Survey Data?

**DOI:** 10.3390/medsci11020037

**Published:** 2023-05-23

**Authors:** Douglas Redd, Terri Elizabeth Workman, Yijun Shao, Yan Cheng, Senait Tekle, Jennifer H. Garvin, Cynthia A. Brandt, Qing Zeng-Treitler

**Affiliations:** 1Center for Data Science and Outcome Research, Washington DC VA Medical Center, Washington, DC 20422, USA; 2Department of Clinical Research and Leadership, George Washington University, Washington, DC 20037, USAstekle@gwu.edu (S.T.); 3VA Salt Lake City Health Care System, Salt Lake City, UT 84148, USA; 4Department of Biomedical Informatics, University of Utah School of Medicine, University of Utah, Salt Lake City, UT 84112, USA; 5VA Connecticut Healthcare System, West Haven, CT 06516, USA; 6Department of Emergency Medicine, Yale School of Medicine, Yale University, New Haven, CT 06510, USA

**Keywords:** natural language processing, dietary supplements, machine learning

## Abstract

There is widespread use of dietary supplements, some prescribed but many taken without a physician’s guidance. There are many potential interactions between supplements and both over-the-counter and prescription medications in ways that are unknown to patients. Structured medical records do not adequately document supplement use; however, unstructured clinical notes often contain extra information on supplements. We studied a group of 377 patients from three healthcare facilities and developed a natural language processing (NLP) tool to detect supplement use. Using surveys of these patients, we investigated the correlation between self-reported supplement use and NLP extractions from the clinical notes. Our model achieved an F1 score of 0.914 for detecting all supplements. Individual supplement detection had a variable correlation with survey responses, ranging from an F1 of 0.83 for calcium to an F1 of 0.39 for folic acid. Our study demonstrated good NLP performance while also finding that self-reported supplement use is not always consistent with the documented use in clinical records.

## 1. Introduction

Dietary supplements are widely used by the general population in the United States. According to the National Health Interview Survey (NHIS), 18% of American adults had used herbal supplements in 2012 [1]. In the 2017–2018 National Health and Nutrition Examination Survey (NHANES), 57.6% of adults reported using dietary supplements in the past 30 days. Women used more dietary supplements than men did across all age categories, and their use increased with age. In particular, adults aged 60 and older reported taking the most supplements, with almost a quarter (24.9%) saying they take four or more [2]. 

However, since dietary supplements are not regulated by the U.S. Food and Drug Administration (FDA) before marketing, the adverse effects and potential drug interactions with prescription medications are not well known. Certain supplements can be dangerous on their own if not taken correctly, and they can also cause adverse events when combined with other substances [3]. In addition, adverse reactions to the products are often not reported [4]. For example, Liu et al. studied the surgical evaluations of 376 patients and found 75% were using complementary therapies, many of which included supplements, but only 17% had discussed it with their physicians [5]. The lack of disclosure has significant implications for patient safety, particularly in situations involving acute care. Patients may unknowingly put their health at risk by taking supplements without proper medical guidance or without considering their potential interactions with other medications. Another study conducted by Wood et al. identified that the use of complementary therapies has become increasingly popular, with a usage rate of 64%. Among these therapies, megadose vitamins and nutritional supplements were found to be the most popular. However, the study also found that many patients did not disclose their complementary use to their healthcare provider [6]. A systematic review of patients with cardiovascular disease found the use of complementary and alternative medicine was common among the patients, with varying prevalence ranging between 19% and 64% of whom were also taking prescription medications. There appeared to be little awareness that there may be interactions with their prescription medications [7]. Many of the known interactions are with prescription cardiac medications, introducing a risk of serious complications [4]. Loya et al. report that 31.5% of their study participants were at risk of experiencing interactions between over-the-counter medications and supplements they were taking [8]. In a Veterans Affairs study, Lee et al. found that 61% of their sample of 200 cancer patients used supplements, and 12% of those were at risk of interactions with prescribed medications [9]. Additionally, a study by Geller et al. found that tens of thousands of emergency room visits every year in the United States are due to adverse events related to dietary supplements [10]. Finally, a study conducted at a geriatrics clinic (*n* = 124) examined the use of dietary supplements and potential interactions with prescription drugs in the elderly population. They identified 23% of participants, and of the supplement users, 54% were at risk for interactions [11]. 

Prevention of adverse events due to these interactions requires knowledge of the medications and supplements used by patients. This is normally obtained from patient medication lists; however, supplements are not routinely present in medication lists unless prescribed by the healthcare provider. Our group was the first to leverage electronic health record (EHR) data in the study of dietary supplements [12,13,14]. We developed a herb-drug interaction alert system that automatically identifies potentially harmful combinations of supplements and drugs from EHR data [12]. In a related study, we found that supplements can be detected more completely through natural language processing (NLP) analysis of patients’ unstructured and semi-structured clinical notes [14]. We were able to identify a significantly larger cohort of individuals using the herbal remedy Ginkgo biloba by analyzing clinical notes rather than relying solely on structured pharmacy data [14]. Specifically, out of a large sample, we were able to identify over 28,000 patients taking Ginkgo biloba based on clinical notes alone, compared to only 9 patients who could be identified through structured pharmacy data. Other researchers have also studied dietary supplements in the context of EHR. Zhang et al. reported that supplement terminology is not fully standardized between medical terminologies and databases. They identified gaps between supplement and standard medication terminologies, making it difficult to identify them in EHR systems. They further found supplements that were not mentioned in the medication list [15]. 

More recently, Arnaud et al. employed NLP techniques to predict the medical specialties of patients at an early stage of hospital admission, integrating structured data with unstructured textual notes from a dataset of about 260 K emergency department records. Their findings show that NLP can accurately predict medical specialties, which has implications for optimizing resource allocation, enhancing patient outcomes, and cutting costs [16]. Additionally, Elbattah et al. reviewed key studies published over the past six years on recent developments and applications of text analytics in healthcare. The main findings highlight the potential of NLP and other text analytics techniques to improve various aspects of healthcare. The study emphasized the need for continued research and development in this field to enhance the effectiveness and efficiency of text analytics in healthcare [17]. Finally, Fan et al. demonstrated the use of rule-based and machine learning models for the detection of supplement use from clinical notes [18], although the study was performed at a single healthcare facility, which may limit generalizability. Additional work on this topic has also been reported [19,20]. 

In this study, we investigated the use of NLP to detect the use of multiple supplements in free-text clinical notes and evaluated the NLP results again using survey data from three different healthcare facilities. Specifically, we applied the trained NLP model to the notes of patients that had completed a survey regarding their supplement use and evaluated the correlation between patients’ self-reported supplement use and the detection of supplement use using NLP techniques on the clinical notes. The aim of this study is to examine the potential of NLP techniques to detect dietary supplement use from both semi-structured and unstructured clinical notes. Our goal is to demonstrate the ability to capture the use of dietary supplements from free text clinical notes and assess the correlation between documented versus self-reported use of dietary supplements. 

## 2. Materials and Methods

### 2.1. Surveys

Paper-based surveys were completed by 377 Veterans from three different Veterans Affairs Medical Centers (VAMCs) (130 from the Washington DC VAMC, 115 from the VA West Haven HCS, and 132 from the VA Salt Lake City HCS) to measure their use of dietary supplements. Informed consent was obtained from each participant after clarification of the study objectives and activities. The characteristics of the participants are shown in Table 1, and a sample of the reported supplements is shown in Table 2. From these surveys, a supplement set was created consisting of the unique supplements used by the participants.

### 2.2. Annotation

This study used medical records from the U.S. Veterans Health Administration (VHA) system under IRB supervision through the VA Informatics and Computing Infrastructure (VINCI). VINCI is a collaboration between the VA Office of Information and Technology (OI&T) and the Office of Research and Development (OR&D) to provide researchers with an environment for secure access to VA healthcare data stored in the Clinical Data Warehouse (CDW) [21]. 

CDW is a national repository for the VA’s electronic health records. It was first “developed in 2006 to accommodate the massive amounts of data being generated from more than 20 years of use and to streamline the process of knowledge discovery and application.” VA pharmacy data are documented in CDW, pharmacy benefits management (PBM), and VHA managerial cost accounting (MCA). Dietary supplements, however, are rarely documented in the structured pharmacy tables [14]. Fortunately, CDW includes unstructured data as textual information utility (TIU) data and houses software applications for NLP. 

A list of keywords was constructed from the supplement set derived from the surveys and augmented with known variants and misspellings. The keywords were grouped into categories when they had the same meaning. These keywords were used to retrieve a set of clinical notes from patients, not including the survey participants, containing the keywords. The documents were tokenized. Relevant punctuation, tabs, and new line markup were not removed prior to annotation review; these elements were operative in the text that was semi-structured. These documents were further split into snippets consisting of the matched keyword and its context, +/−20 words around the keyword. We have found in multiple previous studies that the inclusion of 20 words before and after each keyword results in a more complete context representation than single sentences, partly due to the non-grammatical structures in many clinical notes, which make automatic sentence splitting unreliable. 1000 of these snippets were manually reviewed and annotated to indicate if each keyword occurrence indicated the current use of supplements (yes or no). In snippets containing multiple keywords, all keywords were annotated.

### 2.3. Machine Learning and Evaluation

A support vector machine (SVM) was trained using features derived from the annotated snippets. Each snippet keyword was considered a separate observation, resulting in 1913 observations (many of the 1000 snippets contained multiple keywords). A set of unique bigrams was constructed from all snippets for use as features. The authors used the WEKA workbench [22] in developing the SVM. An additional feature was used to indicate the keyword in the snippet, and the outcome was the supplement annotation (yes or no). The final feature set consisted of 14,217 two-gram features and one keyword feature. A 10-fold cross validation was used to measure performance, and then all observations were used to create a final model.

### 2.4. Survey Evaluation

In order to evaluate the ability of the model to identify supplement use among the survey participants, the model was applied to snippets from clinical notes belonging to the participants. Only clinical notes from the time period between one year prior and one month after the survey date were used in order to maintain context with the surveys. Snippets were retrieved and extracted in the same way as in the training of the SVM, using the supplement keywords, resulting in 28,897 snippets. The SVM model was then applied to those snippets to identify if they indicated active supplement use. If the model predicted active supplement use, then the snippet was classified as positive for the use of the supplement indicated by the keyword. Snippets were then aggregated to the patient level. A patient was considered to be using the supplement if they had at least one snippet positive for the supplement. The methods are illustrated in Figure 1.

## 3. Results

Eighty-three keywords representing 44 categories of dietary supplements were identified (Table 3). The SVM trained on the observations derived from the snippets performed with precision = 0.914, recall = 0.914, and f-measure = 0.914 (10-fold cross validation, details in Table 4).

The snippets (with personal health information removed) in Table 5 are typical of the unstructured and semi-structured data used to train the SVM. The positive keyword examples are in green text, and the negative keyword examples are in orange text. While a central keyword was used to identify the text span, several snippets included multiple keywords, all of which were annotated and used in training. Those classified as “No” sometimes reference allergies, instructional alerts, general food items, laboratory values, or occur as part of a different drug or substance. The examples in Table 5 also highlight various semi-structural elements in the notes, such as enumerated lists, the use of asterisks, and all-caps.

The agreement of the SVM predictions with supplement use self-reported by patients from surveys was calculated by the confusion matrix. Overall, the micro-average precision was 0.79, recall was 0.59, and F1 was 0.68. It was observed that roughly half of the supplements had very low representation (occurring in less than 10% of participants). When excluding those rare supplements, overall agreement was similar: micro-average precision was 0.79, recall was 0.60, and F1 was 0.68 (Table 6). The best agreement of the well-represented supplements, as assessed by the F1 measure, was found in calcium (precision 0.81, recall 0.85, F1 0.83), while the worst was found in folic acid (precision 0.70, recall 0.27, F1 0.39).

## 4. Discussion

Significance: The use of dietary supplements is widespread, with many individuals taking them without a physician’s guidance. However, dietary supplements can potentially interact with both over-the-counter and prescription medications, and many of these interactions are unknown to patients. The documentation of dietary supplement use in structured medical records is often missing, but unstructured clinical notes contain additional information about supplement use. To take advantage of the clinical notes, we developed an NLP system to detect dietary supplement use in clinical notes and used survey data from a group of 377 patients from three healthcare facilities for evaluation.

In particular, we applied the trained model to notes belonging to patients that had completed a survey evaluating their supplement use and evaluated the correlations between patients’ self-reported supplement use and that detected with NLP techniques in the clinical notes. We trained an SVM model on bigrams from snippets of text containing supplement keywords, resulting in an overall high performance (F1 = 0.914). The NLP results’ agreement with patient self-reported surveys, however, was variable, with good agreement in many cases (e.g., calcium, F1 = 0.83) and low agreement in others (e.g., folic acid, F1 = 0.39). 

Implication: Health record systems generally contain medication lists; however, dietary supplements are not routinely tracked via this mechanism. Some dietary supplements have been shown, and others are strongly suspected, of having interactions with other medications. The good news is that we have demonstrated that NLP of semi-structured and unstructured clinical notes can reliably detect the use of many dietary supplements. The not-so-good news is that the agreement between the self-reported supplement use and the NLP results is not consistently high.

Our study has several strengths, including the ability to capture dietary supplement use from free-text clinical notes, which could enable future clinical studies on drug interactions and outcomes research. Moreover, we show that patients from multiple healthcare facilities often self-reported supplement use that contradicted what was recorded in the clinical record, indicating the importance of improving the documentation of supplement use in medical records. Finally, the use of a nationwide electronic health record system allowed for the generalizability of our findings across different healthcare facilities.

Limitations: Despite the promising results of this study, a limitation we encountered was the variability in performance between different supplements when compared to patient self-report in surveys. In error-checking our process, we manually reviewed cases where the SVM prediction did not match the survey answers. This uncovered multiple cases where the patient reported that they were not taking a specific supplement; however, the supplement was recorded in their medication list. There was an apparent pattern indicating that patients did not regard something as a supplement if it was prescribed by a physician. This may be the underlying cause of the lower agreement of our model in cases such as folic acid and melatonin, both of which are commonly prescribed by a physician. Future studies could address the mental model patients have of what they consider to be a supplement. In addition, roughly half of the supplements we studied did not have the number of observations needed for reliable results to be measured when compared to surveys. A larger survey set, or a more directed survey, would be useful to address this. Another limitation is that we did not perform feature selection or parameter optimization before training the SVM, nor did we compare the SVM against other machine learning algorithms. Although SVMs perform their own internal feature selection, better optimization may be obtained from feature selection and parameter tuning.

Another challenge we face in the study of dietary supplements is the lack of dosage, duration, and an exact start date for supplement use. While we were able to identify numerous mentions of the use of dietary supplements, the documentation of dosage, duration, and exact start date is inconsistent, if available at all. Unfortunately, this is a documentation problem that NLP cannot solve. With increasing awareness of the importance of holistic care for patients, we expect the documentation of dietary supplements to improve.

In order to further pursue this important topic, future work will possibly include a larger study with more data and survey participants. Additionally, future work will explore other machine learning algorithms, including the application of large language models such as those produced by bidirectional encoder representations from transformers (BERT), to improve NLP performance.

## 5. Conclusions

In conclusion, we have demonstrated the ability to capture the use of dietary supplements using NLP techniques and found that the NLP results do not always agree with self-reported use in survey data. Our findings underscore the importance of supplement documentation and highlight the need for improved practices to ensure that clinicians have access to accurate and complete information about their patients’ dietary supplement use. Furthermore, the widespread use of dietary supplements and the potential interaction with other medications highlight the importance of providing holistic care for patients, taking into consideration their lives outside of clinical encounters. 

## Figures and Tables

**Figure 1 medsci-11-00037-f001:**
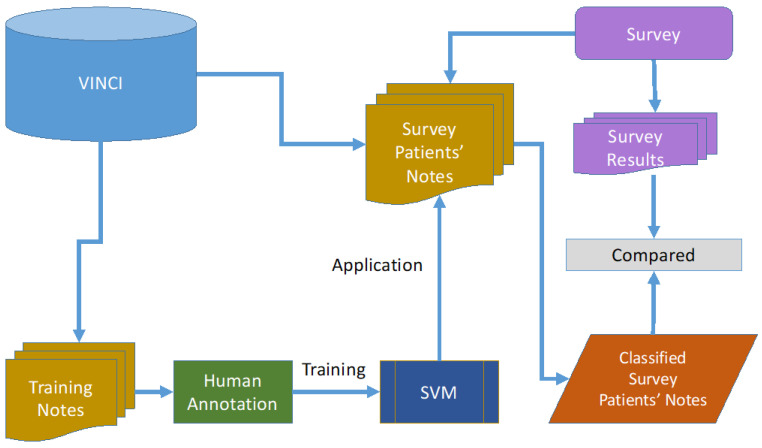
Methods. Notes from other patients are used to train the SVM. Notes from the survey patients are then classified on supplement use; classification output is then compared to the survey results.

**Table 1 medsci-11-00037-t001:** Characteristics of the survey participants.

	Washington DC VA Medical Center	VA West Haven Healthcare System	VA Salt Lake City Healthcare System
Age:	32–88 (61.5)	32–88 (59.1)	24–93 (64.7)
Gender			
Female:	20 (15%)	13 (11%)	7 (5%)
Male:	110 (85%)	101 (88%)	125 (95%)
Race			
Black:	105 (81%)	41 (36%)	9 (7%)
White:	16 (12%)	67 (58%)	115 (87%)
Other:	1 (1%)	2 (2%)	8 (6%)
Unknown: Ethnicity	8 (6%)	5 (4%)	0 (0%)
Hispanic:	1 (1%)	3 (3%)	9 (7%)
Non-Hispanic:	123 (94%)	107 (93%)	123 (93%)
Unknown:	6 (5%)	5 (4%)	0 (0 %)

**Table 2 medsci-11-00037-t002:** Sample of supplement use from surveys.

	Washington DC VA Medical Center	VA West Haven Healthcare System	VA Salt Lake City Healthcare System
Aloe Vera	13	0	0
Black Cohosh	2	0	0
Calcium	0	7	7
Chondroitin	2	8	7
Cinnamon	11	0	2
Coenzyme Q	2	6	7
Cranberry	18	0	0
DHEA	1	1	5
Echinacea	3	2	2
Evening Primrose	1	0	0
Fish Oil	31	13	9
Flaxseed	1	1	1
Folic Acid	10	1	6
Garlic	23	15	7
Ginger	13	1	1
Ginkgo Biloba	7	6	10
Ginseng	10	6	11
Glucosamine	3	17	10
Goldenseal	3	0	0
Green Tea	41	1	2
Iron	0	3	5
Magnesium	0	8	4
Melatonin	3	15	22
Multivitamins	45	49	49
Potassium	0	2	1
Protein	0	1	5
Saw Palmetto	2	2	2
St Johnś wort	1	1	1
Turmeric	6	1	2
Vitamin A	0	1	3
Vitamin B	27	22	32
Vitamin C	30	9	10
Vitamin D	40	27	39
Vitamin E	21	7	2
Yohimbe	2	0	0
Zinc	0	1	2

**Table 3 medsci-11-00037-t003:** Keywords and categories of dietary supplements.

**keywords**	acidophilus, aloe, astragalus, biotin, calcium, cholecalciferol, chondroitin, cinnamon, coenzyme, cranberry, crea, creat, creatine, creatinin, creatinine, creatining, creatnine, cvitamin, cyancobalamin, cyanocobalamin, cyanocobalamine, cyanocobalmin, dhea, digitalis, echinacea, ergocalciferol, fe, ferr, fiber, fish, fishoil, flex, flexion, flx, folic, garlic, ginger, ginkgo, ginko, ginseng, glucosamine, irn, iron, krill, licorice, magnesium, melatonin, mlfolic, multivitamin, multivitamins, multivits, niacin, oregano, palmetto, potasium, potass, potassium, potassuim, pottasium, pyridoxine, retinal, riboflavin, tea, thaimine, thiamin, thiamine, thistle, tumeric, turmeric, vera, viamin, viatmin, vit, vita, vitamin, vitamine, vitd, vitmain, vits, willow, yeast, zinc, zn
**categories**	aloe vera, astragalus, biotin, calcium, chondroitin, cinnamon, coenzyme Q10, cranberry, creatine, curcumin, dhea, digitalis, echinacea, fiber, fish oil, folic acid, garlic, ginger, ginkgo biloba, ginseng, glucosamine, iron, licorice, magnesium, melatonin, multivitamin, oregano, palmetto, potassium, probiotic, tea, thiamine, thistle, vitamin A, vitamin B12, vitamin B2, vitamin B3, vitamin B6, vitamin C, vitamin D2, vitamin D3, willow, yeast, zinc

**Table 4 medsci-11-00037-t004:** Performance of SVM model (10-fold cross validation).

Class	Precision	Recall	F1
Yes	0.929	0.928	0.929
No	0.892	0.894	0.893
Weighted Avg.	0.914	0.914	0.914

**Table 5 medsci-11-00037-t005:** Snippet examples; positive keywords in green text and negative keywords in orange text.

Snippet	Keywords	Class
(14) INSULIN,GLARGINE,HMN 100 UNT/ML INJ ACTIVE Give: 12 UNITS SC QDAY (15) LACTOBACILLUS ACIDOPHILUS/SPOROGENES TAB ACTIVE Give: 1 TABLETS PO TID (16) MELATONIN 5MG SL CAP/TAB ACTIVE Give: 5MG SL	acidophilus, melatonin	Yes (all)
sulfate 325 mg po qday with food *Folic acid 1 mg po qday *Glargine insulin 45 units SC qday *Lactobacillus acidophilus/sporogenes 1 tab po TID *Melatonin 5 mg po SL QHS prn sleep/insomnia *Omeprazole 20 mg po BID AC	folic acid, acidophilus, melatonin	Yes (all)
10.9 and 34.3. Platelets normal at 198,000. Chemistry shows a glucose of 164, chloride 109, and normal creatinine of 0.97. Calcium is 8.1. Protein 6.2. Blood alcohol level is at 284.2. TSH is within normal limits. Acetaminophen and salicylate levels are	calcium, creatinine	No (all)
250 ML expires: 08/12/2016 IV STD PROTOCOL NO LOADING DOSE@1 Instructions too long. See order details for full text. ATORVASTATIN CALCIUM 40MG TAB Give: 40MG PO QPM *INSULIN,ASPART 100UN/ML VIAL 10ML INJ Give: PER PROTOCOL SC QID-INSULIN	calcium	No
Steroid injection helped x 6-8 weeks. Does not take anything for pain. Osteopenia. Dexa 2016. Followed by BHT. Currently on cholecalciferol. PTSD. Having some anger issues. Feels frustrated more easily. Wants to go up on his fluoxetine dose. Says he has	cholecalciferol	Yes
100 mg po q day and triameterene 37.5/HCTZ 25 mg po q day Diabetes: Lantus 30 units twice daily, drinking cranberry juice lately for uti, hasn’t been checking blood sugars Only smoked short time as a teenager PMH: Adenomatous polyp of	cranberry	Yes
Veteran actively engaged in the Therapeutic Lunch Group today from 1130 to noon. 30 veterans shared a lunch of baked fish, french fries, and mixed vegetables. Lunch was planned, prepared, served and cleaned up by veterans. Besides lunch this is an	fish	No
obstruction in the subclavian vessel. Subsequently attempted placement of non-cut 4.5Fr. SL PICC, hoping that the presence of the tapered flex tip on the catheter would be able to overcome the narrowing. This was also unsuccessful. Finally after 40 minutes of	flex	No
-broth or strained broth-based soups -popsicles without pieces of fruit or fruit pulp -water -clear sodas, such as ginger ale, Sprite, or 7-up -sports drinks Try to choose 3 to 5 different varieties of clear liquids for each meal.	ginger	No
DYSPHAGIA, PHARYNGEAL PHASE Insomnia, unspecified (ICD-9-CM 780 Ch DVT/Embl Low Ext Therap Drug Monitor Celiac Disease * (ICD-9-CM 579.0/57 Iron Deficiency Anemia Hypoventilation * (ICD-9-CM 786.09) Gastroesophageal Reflux Disorder * CHRONC PERIODONTITIS NOS Anticoagulants	iron	No
of previous experiences with sedation or analgesia: none REVIEW OF SYSTEM (pertinent to procedure): as per HPI ALLERGIES: SULFA DRUGS, NIACIN CURRENT MEDICATIONS: Active Outpatient Medications (including Supplies): ACCU-CHEK AVIVA PLUS(GLUCOSE) TEST STRIP USE 1 STRIP FOR ACTIVE TESTING BLOOD GLUCOSE	niacin	No

**Table 6 medsci-11-00037-t006:** Performance of SVM model compared to patient reported supplement use from surveys. Results are shown for supplements reported by >10% of participants.

Supplement Category	Precision	Recall	F1
Calcium	0.81	0.84	0.83
Chondroitin	0.78	0.64	0.70
Creatine	0.77	0.76	0.77
Fiber	0.80	0.67	0.73
fish oil	0.79	0.50	0.61
folic acid	0.70	0.27	0.39
Iron	0.81	0.68	0.74
Magnesium	0.81	0.67	0.73
Melatonin	0.68	0.50	0.58
Multivitamin	0.91	0.26	0.40
Potassium	0.76	0.67	0.71
Tea	0.72	0.66	0.69
Thiamine	0.78	0.33	0.47
vitamin A	0.77	0.65	0.71
vitamin B12	0.94	0.28	0.43
vitamin D2	0.88	0.47	0.61
vitamin D3	0.94	0.28	0.43
micro-avg	0.79	0.60	0.68
macro-avg	0.80	0.54	0.62

## Data Availability

The datasets generated during and/or analyzed during the current study are not publicly available to protect the privacy of research participants, but aggregated datasets are available from the corresponding author on reasonable requests.

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
