# Peer review of "Patient Dietary Supplements Use: Do Results from Natural Language Processing of Clinical Notes Agree with Survey Data?"

_medsci, 2023, doi:10.3390/medsci11020037_

Round 1

Reviewer 1 Report

This work is well within the scope of Medical Sciences, and it may be of interest to most of the readers of this journal. It was clear that the main contribution of this work was to demonstrate the ability to capture the use of dietary supplements from free-text clinical notes, enabling clinical studies including drug interactions and outcomes research.

The manuscript does not show introductory background material sufficient for someone, not an expert in this area to understand the context and significance of this work, with insufficient references to follow.

Furthermore, where is the discussion section, please?

This research has reached beneficial conclusions that could be summarized as saying that there are many potential interactions of supplements with both over-the-counter and prescription medications many many of these are not known to the patient. The research team has shown that natural language processing (NLP) of semi-structured and unstructured clinical notes can reliably detect the use of many dietary supplements.

First of all, as the manuscript mention, we need a larger survey set and validated training of the SVM.  Otherwise, the conclusion reached by the research team could serve as a reference guide, until they are verified by other surveys with a larger and more balanced sample so that it can be generalized.

A severe weakness of this research is the fact that Turnitin returned a similarity index of 87% including the references and 76% without bibliography and quotes and it is identified at:

a) www.researchsquare.com (Internet Source)

b) assets.researchsquare.com (Internet Source)

c) www.preprints.org (Internet Source)

d) Submitted to Universita del Piemonte Orientale (student paper)

e) mdpi-res.com (Internet Source)

f) Submitted to Universidade de Évora (student paper), etc…

So, as you understand this work cannot be accepted in this form.

Please, since the conclusions are really very worthwhile rewrite the corresponding points of the manuscript and I would be happy to see and review the final text.

In conclusion, this manuscript is NOT OK, according to Turnitin shows high plagiarism. The English in this paper is good, except for some grammatical mistakes across the text that need proofreading.

I am very sorry, but in order to accept the publication you must add a discussion section and the text will have to be changed so that it is genuinely original.

Finally, for all the above, I have opted to recommend Reconsider after major revision.

Author Response

This work is well within the scope of Medical Sciences, and it may be of interest to most of the readers of this journal. It was clear that the main contribution of this work was to demonstrate the ability to capture the use of dietary supplements from free-text clinical notes, enabling clinical studies including drug interactions and outcomes research.

  • The manuscript does not show introductory background material sufficient for someone, not an expert in this area to understand the context and significance of this work, with insufficient references to follow. 
    • Introduction revised with new references added.

  • Furthermore, where is the discussion section, please?
    • We had previously combined the discussion with conclusion. The revised paper has a separate discussion section with additional content.

  • This research has reached beneficial conclusions that could be summarized as saying that there are many potential interactions of supplements with both over-the-counter and prescription medications many many of these are not known to the patient. The research team has shown that natural language processing (NLP) of semi-structured and unstructured clinical notes can reliably detect the use of many dietary supplements.

  • First of all, as the manuscript mention, we need a larger survey set and validated training of the SVM.  Otherwise, the conclusion reached by the research team could serve as a reference guide, until they are verified by other surveys with a larger and more balanced sample so that it can be generalized.
    • This is a good point. We added a paragraph on Future Work in the Discussion that addresses this.

  • A severe weakness of this research is the fact that Turnitin returned a similarity index of 87% including the references and 76% without bibliography and quotes and it is identified at:
    • We emailed the journal editor with an explanation. The work and the text of this manuscript are completely original, since they are the result of our own funded study by the US Department of Veteran Affairs. This paper was made available as a preprint automatically by the MDPI journal. It was also previously submitted to another journal, officially withdrawn, and confirmed, because the first author of the paper passed away. The MDPI editor (Ms. Nevena Uletilovic) ran a plagiarism check and confirmed that there is no plagiarism.

Please, since the conclusions are really very worthwhile rewrite the corresponding points of the manuscript and I would be happy to see and review the final text. In conclusion, this manuscript is NOT OK, according to Turnitin shows high plagiarism. The English in this paper is good, except for some grammatical mistakes across the text that need proofreading. I am very sorry, but in order to accept the publication you must add a discussion section and the text will have to be changed so that it is genuinely original. Finally, for all the above, I have opted to recommend Reconsider after major revision.

Reviewer 2 Report

I would like to thank the authors for this contribution. The study presents an interesting application of NLP in the medical domain. However, please consider the comments below in the next version.

(1)

While the general topic of the study may be clear, there is a need for additional clarification on the specific motivations and goals of the research. By providing more context and detail on these aspects, the authors can help readers to understand the relevance and the potential implications of the study's findings.

For example, does the literature lack such studies?, or perhaps the present study explores a new application of NLP?

(2)

In the introduction, there is a dire need to refer to further and recent contributions that applied NLP to extract knowledge from free-text clinical notes. For example:

https://doi.org/10.1109/ICHI52183.2021.00103

(3)

Please provide more details on the process of cleaning and preprocessing the data, and any challenges or issues encountered during the data acquisition process  as well.

(4)

I recommend including a figure that illustrates the methodology applied to make it easier for readers to understand the process.

(5)

To ensure reproducibility and transparency, please mention clearly the libraries used to develop the Machine Learning models and cite their references properly.

(6)

Please provide further elaboration on the possible limitations of the study results to give readers a more complete understanding of the study's findings.

(7)

I find that the title is relatively vague. I recommend revising the title to give a more specific aspect about the utilization of NLP.

(8)

Please revise the first paragraph of the introduction, as it appears to have been mistakenly copied from a template.

(9)

Given the growing adoption of transformer models (e.g., BERT) in medical or healthcare research, the authors may wish to consider exploring this area as a potential future direction for their study. Including this point as part of their future work may be beneficial.

Overall, I appreciate the authors' efforts and look forward to seeing an improved version of this study.

Author Response

I would like to thank the authors for this contribution. The study presents an interesting application of NLP in the medical domain. However, please consider the comments below in the next version.

  • While the general topic of the study may be clear, there is a need for additional clarification on the specific motivations and goals of the research. By providing more context and detail on these aspects, the authors can help readers to understand the relevance and the potential implications of the study's findings. For example, does the literature lack such studies? or perhaps the present study explores a new application of NLP?
    • We have revised the literature review and the introduction section in general. Our study demonstrated good NLP performance, while also finding that self-reported supplement use are not always consistent with the documented use in clinical records.

  • In the introduction, there is a dire need to refer to further and recent contributions that applied NLP to extract knowledge from free-text clinical notes. For example: https://doi.org/10.1109/ICHI52183.2021.00103
    • We added more NLP references.
  • Please provide more details on the process of cleaning and preprocessing the data, and any challenges or issues encountered during the data acquisition process as well.
    • We Added more detailed

  • I recommend including a figure that illustrates the methodology applied to make it easier for readers to understand the process.

  • Figure 1 was added to illustrate methods

  • To ensure reproducibility and transparency, please mention clearly the libraries used to develop the Machine Learning models and cite their references properly.

  • This information has been added, with the proper citation.

  • Please provide further elaboration on the possible limitations of the study results to give readers a more complete understanding of the study's findings.
    • The revised limitation section was expanded.

  • I find that the title is relatively vague. I recommend revising the title to give a more specific aspect about the utilization of NLP.

  • We revised the title.

  • Please revise the first paragraph of the introduction, as it appears to have been mistakenly copied from a template.
    • We removed the template text that was included by mistake.

  • Given the growing adoption of transformer models (e.g., BERT) in medical or healthcare research, the authors may wish to consider exploring this area as a potential future direction for their study. Including this point as part of their future work may be beneficial.
    • We added the BERT into the discussion.

Overall, I appreciate the authors' efforts and look forward to seeing an improved version of this study.

Round 2

Reviewer 1 Report

Many thanks to the authors for their efforts to improve the manuscript.

The final result is satisfactory, except that from the third part of the results, you go directly to the fifth part of the discussion without the fourth part of the frost in between which shows a sloppiness. Nevertheless, mistakes are made by accident and for this reason, I would accept the revised version.

Reviewer 2 Report

Thanks for accommodating the feedback, a substantial effort has been put into the revision. I have no further comments.